# Probiotics as Renal Guardians: Modulating Gut Microbiota to Combat Diabetes-Induced Kidney Damage

**DOI:** 10.3390/biology14020122

**Published:** 2025-01-24

**Authors:** Saleh Bakheet Al-Ghamdi

**Affiliations:** Biology Department, Faculty of Science, Al-Baha University, Al-Baha 65779, Saudi Arabia; sb.alghamdi@bu.edu.sa

**Keywords:** gut microbiota modulation, probiotic supplementation, diabetes-induced, renal damage, *Lactobacillus acidophilus*, kidney function preservation, oxidative stress

## Abstract

The gut microbiota plays a crucial role in maintaining health and addressing various diseases, including protecting the kidneys from diabetes-induced damage. This study explored the protective effects of the probiotic *Lactobacillus acidophilus* on kidney health in a rat model of diabetes. We examined four groups: a healthy control group, a probiotic-only group, a diabetic group, and a diabetic group treated with probiotics. Probiotic supplementation began two weeks before diabetes induction, and continued throughout the study. The results revealed that probiotics significantly improved gut health, blood sugar control, and insulin sensitivity in diabetic rats. Probiotic-treated diabetic rats showed restored microbial diversity, improved gut health markers, and reduced oxidative stress. Kidney health also improved, with better tissue structure, reduced inflammation, and less DNA damage observed. These findings suggest that probiotics, by targeting the gut–kidney connection, can be a promising strategy for managing diabetes-related complications and protecting kidney health. This research provides new insights into how natural therapies like probiotics can benefit people with diabetes, but further studies are needed to optimize these treatments and evaluate their long-term effects.

## 1. Introduction

A rising global health concern, diabetes mellitus (DM) is expected to affect 700 million people by 2045. Nearly 40% of cases of end-stage renal disease (ESRD) [1] are also diabetic kidney disease (DKD), one of the most severe consequences of diabetes. Recent research has shown that the start and development of both DM and DKD [2] are much influenced by the gut flora. Comprising trillions of microorganisms, gut bacteria regulate essential processes such as nutrient absorption, immune response, and intestinal barrier integrity [3]. An important therapeutic target for diabetes and DKD [4], dysbiosis—an imbalance in gut microbial composition—has been linked to insulin resistance, systemic inflammation, and renal dysfunction.

Through the synthesis of short-chain fatty acids (SCFAs), most famously butyrate, propionate, and acetate [5], gut bacteria are crucial in controlling glucose metabolism. As signaling molecules, these SCFAs affect insulin sensitivity, energy expenditure, and inflammation [6]. Butyrate, in particular, improves gut barrier integrity by encouraging the expression of tight-junction proteins, lowering gut permeability and preventing systemic inflammation, key elements in lowering insulin resistance [7].

In type 2 diabetes (T2D), dysbiosis is characterized by reduced levels of beneficial bacteria such as Faecalibacterium prausnitzii and Akkermansia muciniphila, which are crucial for anti-inflammatory actions and maintaining gut integrity [8]. The disturbance of these microbial populations increases gut permeability, sometimes referred to as “leaky gut”, allowing bacterial endotoxins like lipopolysaccharides (LPSs) to enter the bloodstream, thus aggravating insulin resistance and hastening metabolic dysfunction [9].

The leading cause of ESRD globally, diabetic kidney disease (DKD) is caused in great part by the gut microbiota and its systemic effects on kidney health; recent studies emphasize this link in pathogenesis [10]. Patients with T2D have been linked to dysbiosis to a higher production of uremic toxins, including p-cresyl sulfate and indoxyl sulfate, which are absorbed into the bloodstream and build-up in the kidneys. These poisons worsen kidney damage by causing oxidative stress, fibrosis, and renal inflammation [11]. In kidney health, gut-derived SCFAs such as butyrate have protective actions by lowering inflammation, oxidative stress, and fibrosis. However, a crucial control of inflammation, nuclear factor kappa-light-chain-enhancement of activated B cells (NF-κB), has been shown to be inhibited by butyrate, reducing renal damage [12]. Furthermore, treatments aimed at gut bacteria, such prebiotics and probiotics, have shown promise in lowering uremic toxin burden and enhancing renal function in diabetic patients [8].

The gut microbiota and its systemic effects on kidney health are a bidirectional link whereby kidney function influences the gut bacteria. Patients with DKD have an altered gut flora which produces uremic toxins, including indoxyl sulfate and p-cresyl sulfate, which aggravate renal inflammation and tubular damage [13]. Made by microbial metabolism, these toxins activate aryl hydrocarbon receptors (AhRs) in renal cells, causing podocyte damage and glomerular sclerosis [14].

Moreover, metabolic dysregulation in diabetes—including hyperglycemia and insulin resistance—exacerbates gut dysbiosis, so aggravating a vicious cycle that hastens renal breakdown. Increased gut permeability connected with hyperglycemia has let bacterial endotoxins enter the bloodstream and cause inflammatory reactions, so aggravating kidney damage [2]. Thus, the pathogenesis of DKD [15] is much influenced by the interaction among gut dysbiosis, uremic toxins, and systemic inflammation.

The critical role of gut microbiota in the progression of diabetic kidney disease (DKD) has spurred significant interest in therapeutic strategies aimed at restoring microbial homeostasis. Interventions such as prebiotics, synbiotics, and probiotics have shown potential in modulating gut microbiota composition and reducing the production of uremic toxins. Probiotic strains, particularly those from the *Lactobacillus* and *Bifidobacterium* genera, have demonstrated efficacy in enhancing gut barrier integrity, decreasing systemic inflammation, and improving renal function in patients with DKD [16]. Additionally, probiotic supplementation has been associated with increased antioxidant capacity and reductions in key markers of oxidative stress and inflammation, including high-sensitivity C-reactive protein (hs-CRP) and malondialdehyde (MDA) [17]. These findings highlight the therapeutic potential of probiotics in mitigating DKD-related complications through modulation of the gut microbiota and its systemic effects on kidney health.

Additionally, showing promise in enhancing metabolic and renal health are prebiotics, which foster the growth of helpful bacteria. Fructo-oligosaccharides (FOSs), for example, lower endotoxemia and increase the abundance of SCFA-producing bacteria, enhancing glucose tolerance and insulin sensitivity in diabetic mice [18]. Combining probiotics and prebiotics, synbiotics have shown benefits in kidney function and lower uremic toxin levels in patients on hemodialysis [19].

The research objective of this study was to investigate the therapeutic potential of modulating gut microbiota through probiotics, specifically *Lactobacillus acidophilus*, in mitigating dysbiosis-induced renal damage associated with diabetic kidney disease (DKD). By analyzing the effects of probiotic supplementation on microbial diversity, pro-inflammatory metabolites, and uremic toxins, this study aimed to provide insights into the interplay between gut health and renal function. The findings will contribute to understanding the long-term implications of microbiota-targeted therapies for improving metabolic health and reducing DKD complications.

## 2. Materials and Methods

### 2.1. Experimental Framework and Study Design

The objective of this study was to evaluate the effects of *Lactobacillus acidophilus* ATCC 4356 (American Type Culture Collection, Manassas, VA, USA) on renal damage induced by diabetes in Sprague–Dawley rats. A total of 40 six-month-old male rats, each weighing between 180 and 200 g, were used and divided equally into four groups (10 rats per group): a control group, a diabetes group, a probiotic-treated group, and a diabetes + probiotic-treated group. The rats were housed under controlled environmental conditions in well-ventilated cages.

Diabetes was induced following a 12 h fast via a single intraperitoneal injection of streptozotocin (STZ) (Sigma-Aldrich, St. Louis, MO, USA) at a dose of 50 mg/kg. Rats with blood glucose levels exceeding 240 mg/dL were classified as diabetic [20,21]. Treatment with *Lactobacillus acidophilus* ATCC 4356 (1 × 10⁹ CFU/kg per day) was administered orally via gavage. The gavaging procedure involved the use of a flexible feeding tube and a calibrated syringe to deliver the probiotic suspension in a 1 mL volume directly to the stomach. Probiotic treatment commenced two weeks prior to the induction of diabetes and continued daily throughout the 12-week experimental period [22,23]. The experimental procedures were reviewed and approved in accordance with the ethical guidelines of Umm Al-Qura University under approval number HAPO-02-K-012-2024-01-1157.

### 2.2. Sample of Blood

The rats’ distal tail vein was used to weekly collect blood samples. The animals were humanely killed after twelve weeks, and blood was drawn using a cardiac puncture. Following separation, the serum was kept at −20 °C for additional biochemical investigations [24].

### 2.3. Biochemical Examinations

Fasting blood glucose (FBG) levels were monitored weekly using a glucometer (Accu-Chek Active, Roche, Basel, Switzerland). Insulin and advanced glycation end-products (AGEs) levels were quantified using ELISA kits (R&D Systems, Minneapolis, MN, USA). The homeostasis model assessment of insulin resistance (HOMA-IR) was calculated using standard equations. Kidney function markers such as blood urea nitrogen (BUN), serum creatinine, and urinary protein levels were measured using commercial kits (BioAssay Systems, Hayward, CA, USA). Oxidative stress markers, including superoxide dismutase (SOD), catalase (CAT), and glutathione peroxidase (GPx), were evaluated using kits (Cayman Chemical, Ann Arbor, MI, USA) according to the manufacturer’s instructions.

### 2.4. Insulin Sensitivity Test (IST) and Oral Glucose Tolerance Test (OGTT)

After a 10 h fast, the oral glucose tolerance test (OGTT) was conducted by administering glucose (Sigma-Aldrich, St. Louis, MO, USA) (2 g/kg) orally. For the insulin sensitivity test (IST), insulin (Novo Nordisk, Bagsværd, Denmark) (0.75 U/kg) was injected intraperitoneally. Blood glucose levels were measured at multiple time points (0, 30, 60, 90, and 120 min) post-administration using a glucometer (Accu-Chek Active, Roche, Basel, Switzerland). The area under the curve (AUC) was calculated using the trapezoidal rule to provide a comprehensive assessment of glucose metabolism [25,26].

### 2.5. Histological Inspection

Kidneys were carefully excised and immediately fixed in 10% neutral buffered formalin (Thermo Fisher Scientific, Waltham, MA, USA) for 24 h. Fixed tissues were dehydrated through a graded alcohol series, embedded in paraffin, and sectioned at a thickness of 5 µm using a microtome (Leica RM2235, Leica Microsystems, Wetzlar, Germany). Sections were stained with hematoxylin and eosin (H&E) to evaluate glomerular and tubular structural integrity. Histological assessments were performed under a light microscope (Olympus CX43, Olympus, Tokyo, Japan) [27].

### 2.6. Comet Assay: DNA Fragmentation

DNA fragmentation, indicative of cellular damage, was assessed using a comet assay kit (Abcam, Cambridge, UK). Tissue homogenates were embedded in low-melting-point agarose on slides and subjected to electrophoresis in an alkaline buffer. The slides were stained with SYBR Green dye (Invitrogen, Waltham, MA, USA) and examined under an epifluorescence microscope (Zeiss Axio Imager, Carl Zeiss, Jena, Germany). Tail moments, a quantitative measure of DNA damage, were analyzed using Comet Assay IV software, version 4.2 (Instem, Staffordshire, UK) [28,29].

### 2.7. Microbiome Analysis

#### 2.7.1. Collection and Treatment of Samples

Treatments were administered orally via gavage once daily for 12 weeks. Initial fecal samples were collected from all groups before treatment commenced. Final fecal samples were collected at the end of the treatment period for microbiome analysis [30].

#### 2.7.2. DNA Extraction from Microorganisms

Fecal DNA was extracted using the Qiagen QIAamp DNA Stool Mini Kit (Qiagen, Hilden, Germany) following the manufacturer’s protocol. DNA quality and concentration were assessed using gel electrophoresis and a NanoDrop 2000 spectrophotometer (Thermo Fisher Scientific, Waltham, MA, USA) [31].

#### 2.7.3. 16S rRNA Gene Sequencing

The V3-V4 region of the 16S rRNA gene was amplified using specific primers:

Forward: 341F (5′-CCTACGGGNGGCWGCAG-3′);

Reverse: 805R (5′-GACTACHVGGGTATCTAATCC-3′).

PCR amplification conditions were as follows:

Initial denaturation: 95 °C for 3 min.

And 30 cycles of the following steps:

Denaturation: 95 °C for 30 s;

Annealing: 55 °C for 30 s;

Extension: 72 °C for 30 s;

Final extension: 72 °C for 5 min.

Amplicons were sequenced using the Illumina MiSeq platform (Illumina Inc., San Diego, CA, USA). Sequencing data were processed and analyzed to identify bacterial taxa [32].

#### 2.7.4. Processing and Filtering of Sequencing Data

Raw sequencing data were processed and filtered using bioinformatics tools, primarily QIIME2 (Quantitative Insights Into Microbial Ecology, version 2022.2). The following quality control steps were included:

Removing low-quality reads;

Eliminating chimeric sequences;

Merging paired-end reads to reconstruct full-length sequences.

Operational taxonomic units (OTUs) were clustered at a 97% similarity threshold. Taxonomic classification of the 16S rRNA gene sequences was performed using reference databases, specifically the SILVA database (version 138.1) and Greengenes database (version 13.8), for an accurate identification of microbial taxa [33,34].

#### 2.7.5. Alpha Diversity Analysis

To evaluate the microbial diversity within each group, alpha diversity indices were calculated:

Chao1 Index: Estimated species richness;

Shannon Diversity Index: Measured total microbial diversity.

All indices were computed using QIIME2, and the results were visualized using bar plots to facilitate comparison among groups [34,35].

#### 2.7.6. Beta Diversity and Principal Component Analysis (PCA)

Beta diversity was analyzed using weighted UniFrac distances to assess the variations in microbial community composition between groups. Principal component analysis (PCA) was conducted to visualize clustering patterns and differences in microbial composition among the control and treatment groups. These analyses provided insights into the impact of *Lactobacillus acidophilus* supplementation on microbial structure [30].

#### 2.7.7. Taxonomic Composition and Heatmap Visualization

Statistical analysis of bacterial taxa at the phylum and genus levels was conducted to determine their relative abundance. Prominent bacterial phyla included the following: Actinobacteria, Firmicutes, Bacteroidetes, Verrucomicrobia, Proteobacteria.

A stacked bar chart was used to display the relative proportions of these phyla across groups. Additionally, a heatmap was generated to illustrate the abundance of key microbial taxa, highlighting the differences in composition among experimental groups [36].

#### 2.7.8. Firmicutes/Bacteroidetes Ratio

The Firmicutes/Bacteroidetes (F/B) ratio, a critical marker of gut health, was calculated for each group. Changes in this ratio were plotted to evaluate the effect of varying dosages of *Lactobacillus acidophilus*. This analysis provided insights into the probiotic’s potential role in restoring microbial balance [36].

### 2.8. Analysis of Statistics

Both R and GraphPad Prism (R Software: utilized version 4.3.1, released in June 2023; GraphPad Prism: utilized version 10.12, released in 2023) were used in the process of conducting all statistical analyses. Comparisons of the alpha diversity (Chao1 and Shannon indices) across the groups were studied using a one-way analysis of variance (ANOVA), and then Tukey’s post hoc test was performed. We used Kruskal–Walli’s test to determine whether or not there were any differences in the taxonomic makeup. In the statistical analysis, a *p*-value that was lower than 0.05 was regarded as statistically significant [37].

## 3. Results

### 3.1. Effect of Lactobacillus acidophilus on Body Weight and Insulin Levels in Diabetic Rats

The impact of *Lactobacillus acidophilus* ATCC 4356 on body weight and insulin levels in diabetic rats was closely monitored throughout the experiment. The results, illustrated in Figure 1A, demonstrate significant changes in body weight among the experimental groups. Following the induction of diabetes via streptozotocin (STZ), the diabetic rats exhibited a marked decrease in body weight compared to the consistent weight gain observed in the control group.

However, treatment with *L. acidophilus* significantly mitigated this weight loss. By the end of the study, the probiotic-treated diabetic group had noticeably higher body weights compared to the untreated diabetic group (*p* < 0.05), indicating the protective effect of *L. acidophilus* against diabetes-induced weight reduction.

Figure 1B highlights the effect of *L. acidophilus* on fasting blood glucose (FBG) levels. Diabetic rats treated with the probiotic showed a significant decrease in FBG levels compared to the untreated diabetic group (*p* < 0.05), further demonstrating the beneficial role of the probiotic in improving glycemic control.

Figure 1A shows the variations in body weight among several rat groups over the experiment. Whereas the control group maintained constant weight gain, the diabetic rats showed a significant drop in body weight following the induction of type 2 diabetes (T2D) using streptozotocin (STZ). *Lactobacillus acidophilus’s* administration helped diabetic rats’ usual body weight loss be lessened. *Lactobacillus acidophilus* clearly lowers body weight loss in T2D rats since, by the end of the study, the body weight of the probiotic-treated group was much higher than that of the untreated T2D group (** p* < 0.05). Figure 1B shows how well *Lactobacillus acidophilus* lowers fasting blood glucose (FBG) levels; this is shown by a notable drop in FBG in the treated rats as compared to the diabetic rats treated differently (* *p* < 0.05).

### 3.2. Impact of Lactobacillus acidophilus on T2D Rat Glucose Tolerance and Insulin Sensitivity

We investigated insulin resistance employing the Homeostasis Model Assessment for Insulin Resistance (HOMA-IR) and β-cell function (HOMA-β) in order to further assess the positive effects of *Lactobacillus acidophilus* in diabetic rats. Additionally performed to evaluate glucose metabolism and insulin sensitivity were the Oral Glucose Tolerance Test (OGTT) and Insulin Sensitivity Test (IST). 

With a *p* < 0.0001, the probiotic-treated group displayed a clear drop in HOMA-IR levels when compared to the T2D group, so indicating enhanced insulin sensitivity. Reflecting improved pancreatic β-cell function, the HOMA-β values in the probiotic-treated group were also considerably higher than those in the untreated T2D group.

The T2D group showed a significantly greater glycemic response than the control rats on both the OGTT and IST tests (Figure 2C,D). On the 120 min observation period, however, the group treated with *Lactobacillus acidophilus* showed noticeably lower blood glucose levels (*p* < 0.01; Figure 2E). These results imply that *Lactobacillus acidophilus* can reduce glucose metabolic abnormalities in diabetic rats, so possibly reducing the risk of complications connected to diabetes including kidney damage.

Figure 2 shows in Type 2 diabetic (T2D) rats the effects of *Lactobacillus acidophilus* on several metabolic parameters. Whereas 2B shows the Homeostatic Model Assessment for Beta Cell Function (HOMA-β), Figure 2A shows the Homeostatic Model Assessment for Insulin Resistance (HOMA-IR). Figure 2C,D, respectively, highlights the area under the curve (AUC) calculations for the Oral Glucose Tolerance Test (OGTT) and Insulin Sensitivity Test (IST). Figure 2E,F also shows week 8 OGTT and IST results. Data are shown as the means ± standard deviation (n = 8–10). Comparative to the T2D group, statistical significance is indicated as follows: ** p* < 0.05, * *p* < 0.001.

### 3.3. Impact of Lactobacillus acidophilus on T2D Rat Antioxidant Enzyme Levels

The effect of *Lactobacillus acidophilus* on antioxidant enzyme levels in rats suffering with type 2 diabetes (T2D) is described in this part. Comparative to the control group, the T2D group displayed a notable decline in antioxidant enzymes, including catalase (CAT), superoxide dismutase (SOD), and glutathione peroxidase (GPx) levels (*p* < 0.01, Figure 3A–G). But by 14.96% (*p* < 0.05), 21.96% (*p* > 0.05), and 15.91% (*p* < 0.01), treatment with *Lactobacillus acidophilus* resulted in increases in SOD, CAT, and GPx levels, respectively. On the other hand, glutathione-S-transferase (GST) activity displayed a significant increase in the diabetic group relative to both the control and *Lactobacillus acidophilus*-treated groups.

Figure 3 shows in type 2 diabetic (T2D) rats the effects of *Lactobacillus acidophilus* on several important biochemical parameters. Data are stated as the means ± standard deviation, with n = 8–10. While * *p* < 0.05 and ** *p* < 0.01 indicate comparisons to the *Lactobacillus acidophilus*-treated group, statistical significance is indicated by *p* < 0.05 and *p* < 0.01 against the T2D group.

### 3.4. Assessment of Renal Health Parameters

The study also assessed *Lactobacillus acidophilus’s* renal protective properties in T2D rats, so underscoring its part in both structural and functional enhancement of the kidneys. Earlier studies have indicated that probiotic supplements help avoid metabolic abnormalities including insulin resistance and dyslipidemia. It has been noted to normalize glucose metabolism, lower adipose tissue, increase glucose and insulin tolerance, and restore liver and kidney functions. These developments point to the possibility of probiotics such as *Lactobacillus acidophilus* lowering renal damage brought on by diabetes.

Compared to untreated diabetic rats in Image (Control—LA), rats treated with *Lactobacillus acidophilus* showed notably lower levels of albumin and glucose in urine samples, indicators of kidney damage in the present work. Along with reversing structural defects in kidney tissues Image (T2D), the probiotic treatment restored renal glucose metabolism markers and fibrosis. Furthermore, *Lactobacillus acidophilus* intake reduced kidney oxidative stress markers, including reactive oxygen species (ROS), which also helped lower triglyceride accumulation.

From a functional standpoint, *Lactobacillus acidophilus* proved able to offset diabetic oxidative damage in Image (T2D+LA). Reduced liver triglyceride levels, enhanced insulin resistance, and lower dyslipidemia all pointed clearly toward this. Treating diabetic rats with *Lactobacillus acidophilus* enhanced kidney performance without appreciable body weight loss, suggesting a possible preventive mechanism against kidney damage.

These results in Figure 4A–C taken together, strongly support the use of *Lactobacillus acidophilus* as a natural therapeutic agent for reducing diabetes-induced renal damage by means of better renal structure and function, and enhanced antioxidant defense mechanisms.

### 3.5. Comet Assay of Kidney Tissue in T2D Rats

This part centers on the Comet assay, a gel electrophoresis-based technique for evaluating type 2 diabetes (T2DM) DNA damage in kidney cells from rats. The assay is well-known for detecting single- and double-stranded breaks, crosslinks, and apoptotic nuclei, among other types of DNA damage. Its sensitivity lets one detect almost 50 strand breaks per diploid mammalian cell.

Designed first to assess DNA damage and repair systems in mammalian cells, the test takes 24 h to complete. Examining kidney tissue, the T2DM group showed notable DNA damage; rats treated with *Lactobacillus acidophilus* (T2D+LA) showed a marked decrease in DNA fragmentation (Figure 5, A, C, C+LA, T2D, T2D+LA). This effect implies that *Lactobacillus acidophilus* reduces DNA damage caused by diabetes in kidney cells.

X-rays on ice were used to expose single-cell suspensions, reducing strand break rejoining during the assay. After varying radiation levels 2 Gy, 30 Gy, 8 Gy, and 60 Gy, the results show relative outcomes for the T2D and T2D+LA groups. Figure 5 shows dose–response curves, showing the relationship between radiation exposure and apoptotic tail length in kidney cells of every group (mean ± SD; n = 10).

Reduced tail length in the comet assay for the T2D+LA group across different radiation levels shows that *Lactobacillus acidophilus* treatment considerably decreased DNA damage in kidney cells when compared to untreated T2D rats.

### 3.6. The Constituents of the Microbiome

#### 3.6.1. Bacterial Phyla Proportions

Analysis of the bacterial phyla composition revealed significant differences among the experimental groups. The control group exhibited the highest abundance of *Firmicutes* (80%) and a smaller proportion of *Bacteroidetes* (10%), with minor contributions from other phyla. In contrast, the T2D group showed a drastic reduction in *Firmicutes* (30%) and an increase in *Bacteroidetes* (30%), indicative of severe dysbiosis. Supplementation with *Lactobacillus acidophilus* in the T2D+*L. acidophilus* group partially restored the microbial balance, with *Firmicutes* increasing to 50%.

#### 3.6.2. Shannon Diversity Index

The Shannon diversity index, a measure of microbial diversity, demonstrated the highest values in the control group (4.5 ± 0.1), reflecting a stable and diverse microbiota. Conversely, the T2D group showed the lowest diversity (2.5 ± 0.3), indicative of disrupted microbiota. Treatment with *L. acidophilus* in the T2D+*L. acidophilus* group significantly improved microbial diversity (3.8 ± 0.15), demonstrating partial recovery of the gut microbiota composition.

#### 3.6.3. Firmicutes/Bacteroidetes (F/B) Ratio

The *Firmicutes/Bacteroidetes* (F/B) ratio, a key indicator of gut microbiome health, was highest in the control group (2.0 ± 0.1). The T2D group exhibited a significant reduction in the ratio (0.8 ± 0.2), indicative of microbial imbalance commonly associated with metabolic disorders. *L. acidophilus* supplementation restored the F/B ratio to 1.5 ± 0.12 in the T2D+*L. acidophilus* group, suggesting improved microbial composition and gut health.

#### 3.6.4. Heatmap Analysis of Bacterial Abundance

Heatmap analysis showed stark differences in bacterial abundance between groups. *Firmicutes* dominated the control group (80%), but their abundance was substantially reduced in the T2D group (30%), with an accompanying increase in *Bacteroidetes*. In the T2D+*L. acidophilus* group, supplementation partially restored *Firmicutes* levels to 50%, underscoring the ameliorative potential of probiotics.

#### 3.6.5. Principal Coordinate Analysis (PCoA)

Principal coordinate analysis (PCoA) revealed distinct clustering patterns among the microbial communities of the groups. The control group formed a tight cluster, indicative of a stable microbiome, whereas the T2D group displayed dispersed clustering, reflecting significant microbial disruption. Treatment with *L. acidophilus* shifted the clustering of the T2D+*L. acidophilus* group closer to the control cluster, suggesting partial restoration of microbial stability.

#### 3.6.6. Microbial Characteristics

Microbial characteristic analysis highlighted significant differences among groups. The control group consistently exhibited higher levels of beneficial microbial features compared to the T2D group. Supplementation with *L. acidophilus* improved these features in the T2D+*L. acidophilus* group, particularly in metrics associated with bacterial diversity and gut health, approaching values observed in the control group.

#### 3.6.7. Firmicutes and Bacteroidetes Proportions

In the control group, *Firmicutes* accounted for 80% of the total microbiota, compared to 20% for *Bacteroidetes*, reflecting a healthy microbiota balance. In the T2D group, *Firmicutes* proportions dropped to 40%, while *Bacteroidetes* increased to 60%, indicating dysbiosis. Supplementation with *L*. *acidophilus* in the T2D+*L*. *acidophilus* group partially restored the balance, with *Firmicutes* increasing to 60% and *Bacteroidetes* decreasing to 40%.

The bar chart represents the relative abundance of bacterial phyla across the groups. The control group exhibited the highest proportion of Phylum 1 (80%), followed by Phylum 2 (10%), with minimal contributions from Phylum 3 and Phylum 4. In contrast, the T2D group demonstrated a reduction in Phylum 1 (30%) and an increase in Phylum 2 (30%) and other phyla, indicative of microbiome dysbiosis. Supplementation with *L. acidophilus* in the T2D+*L*. *acidophilus* group partially restored these proportions, with Phylum 1 increasing to 50%.

The bar chart illustrates the Shannon Diversity Index, a measure of microbial diversity, for each group. The control group recorded the highest diversity (4.5 ± 0.1), reflecting a healthy microbiome. The T2D group displayed the lowest diversity (2.5 ± 0.3), indicating a substantial microbial imbalance. Supplementation with *L*. *acidophilus* in the T2D+*L. acidophilus* group resulted in an improved diversity index (3.8 ± 0.15), suggesting a partial restoration of microbial balance.

The *Firmicutes/Bacteroidetes* ratio, a key indicator of gut health, was substantially improved in the probiotic-treated diabetic group. The T2D group had a significantly reduced ratio (0.8 ± 0.2), consistent with microbial dysbiosis associated with metabolic disorders. The T2D+*L. acidophilus* group showed an improved ratio (1.5 ± 0.12) compared to the untreated diabetic group, indicating a rebalancing effect.

The heatmap displays the relative abundance of bacterial phyla across the groups. Phylum 1 was predominant in the control group (80%), whereas the T2D group showed a marked reduction (30%), with increased contributions from other phyla. The T2D+*L*. *acidophilus* group displayed a partial restoration of Phylum 1 abundance (50%), indicating the positive impact of supplementation. Numerical values corresponding to each cell are overlaid on the heatmap.

The bar chart highlights the proportions of specific microbial characteristics across the groups. The control group displayed consistently higher proportions across all characteristics compared to the T2D group. Supplementation with *L. acidophilus* in the T2D+*L. acidophilus* group resulted in an improvement, particularly in characteristics 1 and 2, with proportions approaching those of the control group.

The bar chart shows the proportions of *Firmicutes* and *Bacteroidetes* in each group. The control group had a higher proportion of *Firmicutes* (80%) compared to *Bacteroidetes* (20%), representing a healthy balance. The T2D group exhibited a disrupted balance, with reduced *Firmicutes* (40%) and increased *Bacteroidetes* (60%). Supplementation with *L. acidophilus* in the T2D+*L. acidophilus* group partially restored the balance, with *Firmicutes* increasing to 60% and *Bacteroidetes* decreasing to 40%.

The principal coordinate analysis (PCoA) plot demonstrates distinct clustering of microbial communities. The control group formed a tight cluster, reflecting microbial stability, while the T2D group showed a dispersed pattern, indicative of microbiome disruption. The T2D+*L*. *acidophilus* group exhibited a shift in clustering closer to the control, suggesting partial recovery of the microbial community structure.

## 4. Discussion

This work emphasizes how *Lactobacillus acidophilus* might be therapeutically beneficial in a type 2 diabetes (T2D) model by enhancing renal function and thus minimizing metabolic disturbances. The results are in line with current studies on gut microbiota and its systemic effects on kidney health and show that modification of gut flora can greatly reduce renal damage brought on by diabetes.

### 4.1. Impact on Insulin Sensitivity and Metabolic Measures

The favorable effects of *Lactobacillus acidophilus* on glucose metabolism and insulin sensitivity observed in this study aligning with findings from previous research. Like other gut-modulating treatments, the lower Homeostasis Model Assessment for Insulin Resistance (HOMA-IR) and higher HOMA-β seen in the probiotic-treated rats point to *Lactobacillus acidophilus* clearly reducing insulin resistance. Particularly butyrate, which strengthens gut barrier integrity and lowers systemic inflammation, these findings align with earlier research emphasizing how probiotics improve glucose homeostasis by SCFA generation [38,39]. Often referred to as “leaky gut”, increased gut permeability in T2D lets endotoxins including lipopolysaccharides (LPS) into the bloodstream, so aggravating inflammation increases insulin resistance and renal damage [40]. The reduced glycemic response in *Lactobacillus acidophilus*-treated rats suggests that probiotics restore gut integrity, preventing the translocation of harmful endotoxins into the bloodstream and reducing systemic inflammation and metabolic dysfunction. These results are in line with studies implying that by restoring the gut microbiota balance, probiotics can change inflammation and glucose tolerance [41].

### 4.2. Protective Effects on Renal Function

Furthermore, implying a protective function for *Lactobacillus acidophilus* is the result on kidney health. Reduced blood urea nitrogen (BUN), serum creatinine, and total urinary protein in treated rats point to a clear improvement in kidney function, which could be ascribed to the anti-inflammatory actions of SCFAs generated by the gut flora. Key SCFA butyrate has been shown to block inflammatory pathways including NF-κB activation, which is vitally important in the course of diabetic kidney disease [7,42]. This is consistent with past research showing that gut-derived SCFAs can lower oxidative stress and fibrosis [6] so mitigating renal damage.

Histological analysis supports the theory that *Lactobacillus acidophilus* mitigates diabetes-induced renal damage by reducing fat deposition, congestion, and inflammation in the kidneys of treated rats. These structural changes line up with earlier research showing that by lowering fibrosis and oxidative stress [43,44], SCFAs strengthen the integrity of the kidney’s cellular architecture.

### 4.3. DNA Protection

The comet assay findings showed that rats given *Lactobacillus acidophilus* had notable decreases in DNA damage in their kidneys. A main feature of oxidative stress, DNA damage is crucial for the development of diabetic complications. The comet assay of the treated group’s shortened tail length points to a probiotic lowering of oxidative stress at the cellular level, so preventing DNA fragmentation. These results align with studies [45,46], demonstrating that probiotic supplementation increases antioxidant enzyme activity, including catalase (CAT), which scavenges reactive oxygen species (ROS) and reduces oxidative damage, and superoxide dismutase (SOD), which scavenges reactive oxygen species (ROS) and so helps.

#### 4.3.1. Comparing Gut Microbiota Modulation Therapies

The general therapeutic advantages shown in this work highlight the need of focusing on the gut flora to control diabetic kidney disease. Like other microbiota-modulating treatments, probiotics including *Lactobacillus acidophilus* show promise in reversing gut dysbiosis, lowering systemically inflammation, and improving renal outcomes [47,48]. Probiotics are a useful complement therapy in controlling diabetes-related complications, including DKD since their ability to affect metabolic health by gut microbiota and its systemic effects on kidney health modulation.

#### 4.3.2. The Role of *Lactobacillus acidophilus* in Modulating Gut Microbiota Dysbiosis in Type 2 Diabetes

The gut microbiota plays a fundamental role in maintaining host metabolic health, and its dysbiosis has been strongly associated with the development and progression of type 2 diabetes (T2D) [49]. This study investigated the impact of *Lactobacillus acidophilus* supplementation on gut microbiota composition in T2D-induced mice. The results revealed significant improvements in microbial diversity, community structure, and metabolic markers, indicating the therapeutic potential of probiotics.

#### 4.3.3. Bacterial Phyla Proportions

Analysis of bacterial phyla proportions (Figure 6) demonstrated significant changes in the T2D group, including a marked reduction in *Firmicutes* and an increase in *Bacteroidetes*. These findings align with prior studies linking an altered *Firmicutes*/*Bacteroidetes* (F/B) ratio to metabolic disorders [50]. Supplementation with *L. acidophilus* partially restored the abundance of *Firmicutes*, highlighting its ability to mitigate gut dysbiosis.

#### 4.3.4. Microbial Diversity

Microbial diversity, assessed using the Shannon diversity index (Figure 7), was significantly reduced in the T2D group, consistent with decreased microbial diversity in diabetic conditions [51]. Treatment with *L. acidophilus* improved diversity levels, though not to the extent observed in the control group. This partial recovery underscores the importance of probiotics in restoring microbial homeostasis, a critical factor for maintaining metabolic health.

#### 4.3.5. *Firmicutes*/*Bacteroidetes* (F/B) Ratio

The F/B ratio, a key indicator of gut microbiota balance, was the lowest in the T2D group (Figure 8), reflecting increased intestinal permeability and systemic inflammation, both hallmarks of T2D [52]. Probiotic supplementation significantly increased the F/B ratio, suggesting a rebalancing of microbial composition and a potential reduction in inflammation and gut barrier dysfunction.

#### 4.3.6. Heatmap and Microbial Characteristics

Heatmap analysis (Figure 9) and microbial characteristics (Figure 10) showed that *L*. *acidophilus* supplementation enhanced specific microbial features associated with gut health. These improvements likely contributed to better metabolic outcomes, as previously reported in similar studies [53].

#### 4.3.7. Restoration of *Firmicutes* and *Bacteroidetes* Proportions

The analysis of *Firmicutes* and *Bacteroidetes* proportions (Figure 11) indicated a significant recovery in the probiotic-treated group. In the T2D group, *Firmicutes* proportions dropped to 40%, while *Bacteroidetes* increased to 60%, indicative of dysbiosis. Probiotic treatment restored the balance, with *Firmicutes* increasing to 60% and *Bacteroidetes* decreasing to 40%, aligning with findings from prior research highlighting the role of probiotics in modulating gut microbiota for improved metabolic health outcomes [54].

#### 4.3.8. Principal Coordinate Analysis (PCoA)

Principal coordinate analysis (Figure 12) revealed clear disruptions in microbial community structure in the T2D group, with clustering patterns distinct from the control group. Following treatment with *L. acidophilus*, the microbiome composition in the probiotic-supplemented group shifted closer to that of the control group, indicating partial restoration of microbial stability. These results are consistent with prior studies showing the ability of probiotics to improve gut microbiota composition in diabetic models [53].

#### 4.3.9. Conclusion

This study demonstrates that *Lactobacillus acidophilus* supplementation effectively mitigates gut microbiota dysbiosis in T2D by restoring microbial diversity, rebalancing key phyla proportions, and improving gut health. These findings suggest that *L. acidophilus* could alleviate metabolic disturbances associated with T2D through gut microbiota modulation. Further research is needed to explore the long-term efficacy and underlying mechanisms of probiotic therapy in managing metabolic disorders.

## 5. Conclusions

The results of this study imply that supplements of *Lactobacillus acidophilus* protect kidney function and enhance metabolic parameters in rats suffering from renal damage caused by diabetes. These findings underline even more the important function of the gut microbiota and its systemic effects on kidney health in diabetic kidney disease and support the therapeutic possibilities of probiotics in controlling complications connected to diabetes. The development in glucose tolerance, insulin sensitivity, antioxidant defense, and renal function emphasizes the possibility of *Lactobacillus acidophilus* to be included into therapeutic plans meant to stop DKD development. Future studies, nevertheless, should investigate the long-term consequences and clinical relevance of such treatments in humans.

## Figures and Tables

**Figure 1 biology-14-00122-f001:**
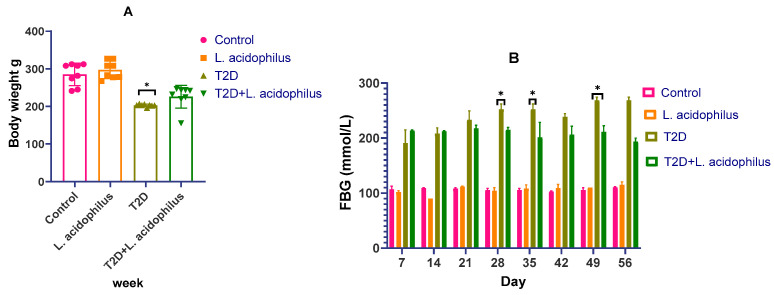
(**A**,**B**) The impact of *Lactobacillus acidophilus* on body weight and insulin levels in diabetic rats. ** p* < 0.05.

**Figure 2 biology-14-00122-f002:**
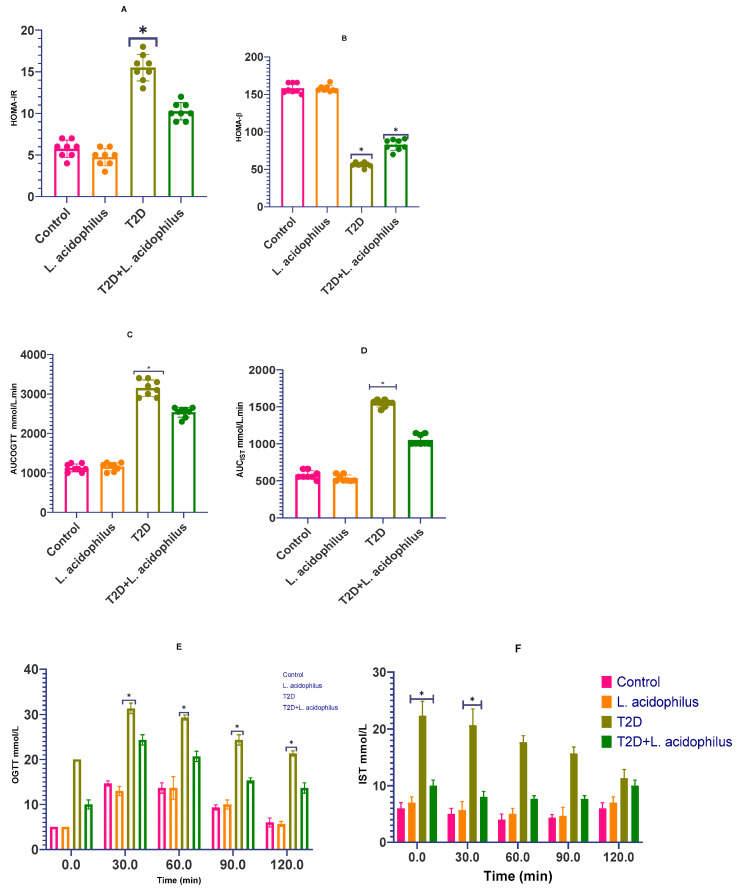
(**A**–**F**) The influence of *Lactobacillus acidophilus* on glucose tolerance and insulin sensitivity in T2D rats. ** p* < 0.05.

**Figure 3 biology-14-00122-f003:**
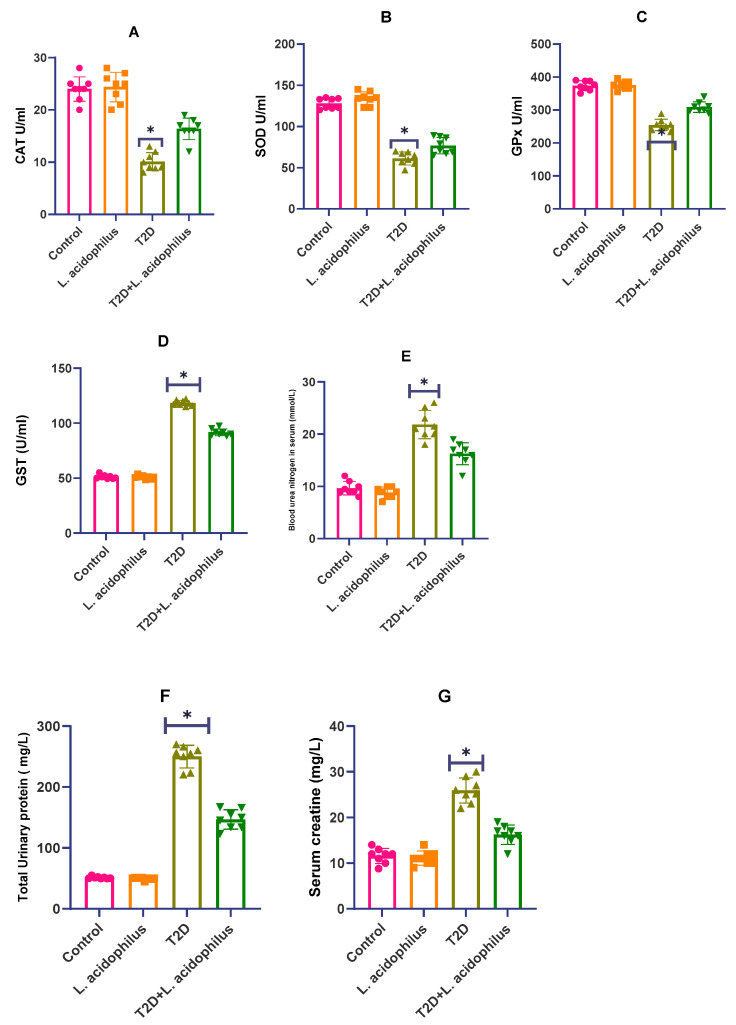
(**A**–**G**) Impact of *Lactobacillus acidophilus* on kidney function and antioxidant markers in T2D rats. ** p* < 0.05.

**Figure 4 biology-14-00122-f004:**
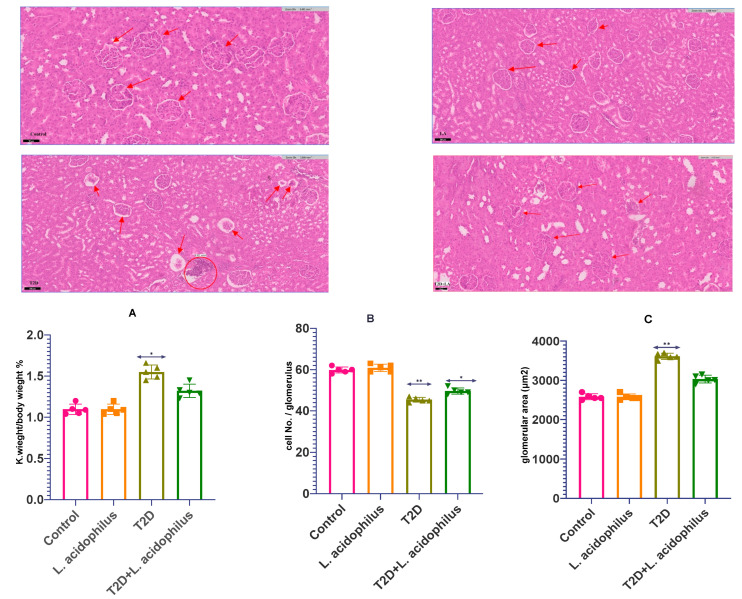
(Image (control—LA—T2D—T2D+LA) (**A**–**C**)) This figure shows how *Lactobacillus acidophilus* supplementation protects kidneys against tissue damage in type 2 diabetes mellitus (T2DM) rats. Control (C), *Lactobacillus acidophilus*, T2DM treated (T2D), and T2DM treated with *Lactobacillus acidophilus* (T2D+LA) are among the groups shown. Comparative to the normal control group, statistical significance is indicated as *p* < 0.05. Furthermore, mentioned against the diabetic control group are * *p* < 0.05 and ** *p* < 0.01. Image 1 (Control) in Image (Control–T2D+LA) microscopic images derived from hematoxylin and eosin (H&E)-stained kidney sections. The kidney tissue looks normal. There is no evidence of damage or inflammation among the well-defined and orderly glomeruli. This picture stands for the control or healthy condition. Image 2: (C+LA) This picture resembles the first one (control) rather exactly. The glomeruli or surrounding tissue are not clearly changed or damaged. One could regard it as either normal or almost normal. Third image (T2D): This picture clearly exhibits pathogenic changes. An accumulation of inflammatory cells in the circular area points to the fibrosis or inflammation there. Furthermore, changing in the surrounding tissue are indications of more severe kidney damage than in the first two pictures. Fourth image (T2D+LA): Less dramatic changes are shown in this picture than in Image 3. Though there are still some indications of enlarged intercellular spaces and mild degenerative changes, the glomeruli seem in better condition with better tissue organization. Though not totally normal, this condition is better than in Image 3. Having n = 10 for every group, the data are shown as the median interquartile range (IQR). These results highlight, by lowering structural damage in diabetic kidney tissues, the possible kidney-protective action of *Lactobacillus acidophilus*.

**Figure 5 biology-14-00122-f005:**
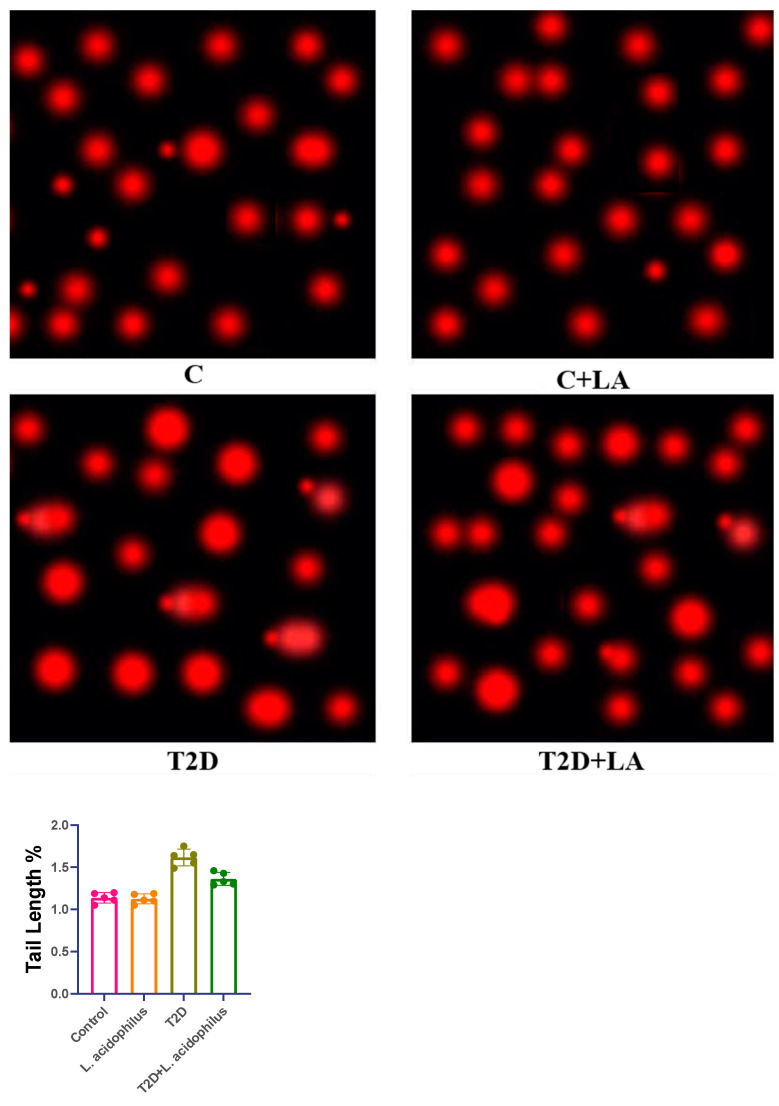
Four separate groups of rats—the control group, the type 2 diabetes (T2D) group, and the type 2 diabetes with *Lactobacillus acidophilus* treatment (T2D+LA) group—have their tail lengths (µm) shown here. The experimental technique used the overnight alkaline comet assay, which is good in identifying DNA single-stranded breaks, double-stranded breaks, and alkali-labile sites.

**Figure 6 biology-14-00122-f006:**
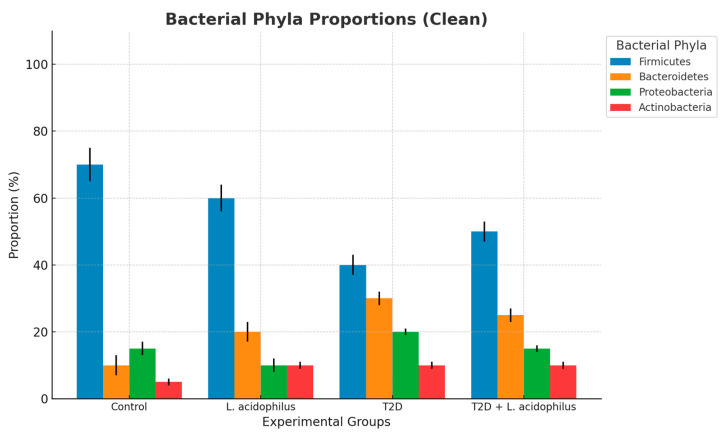
Bacterial phyla proportions.

**Figure 7 biology-14-00122-f007:**
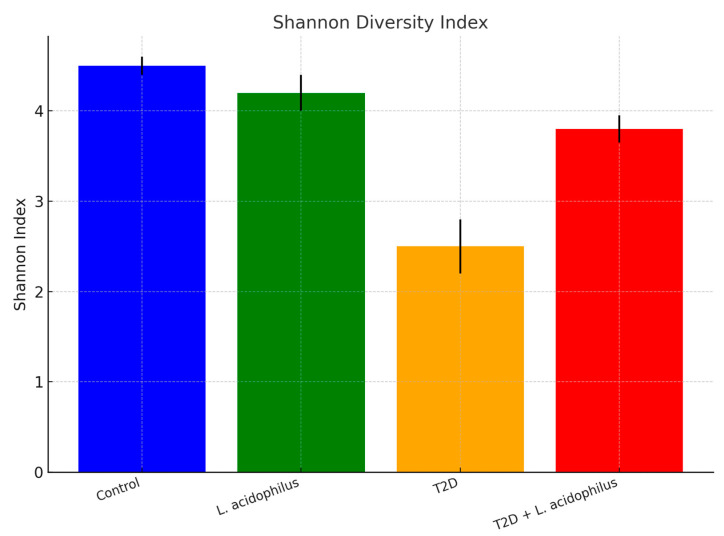
Shannon diversity index.

**Figure 8 biology-14-00122-f008:**
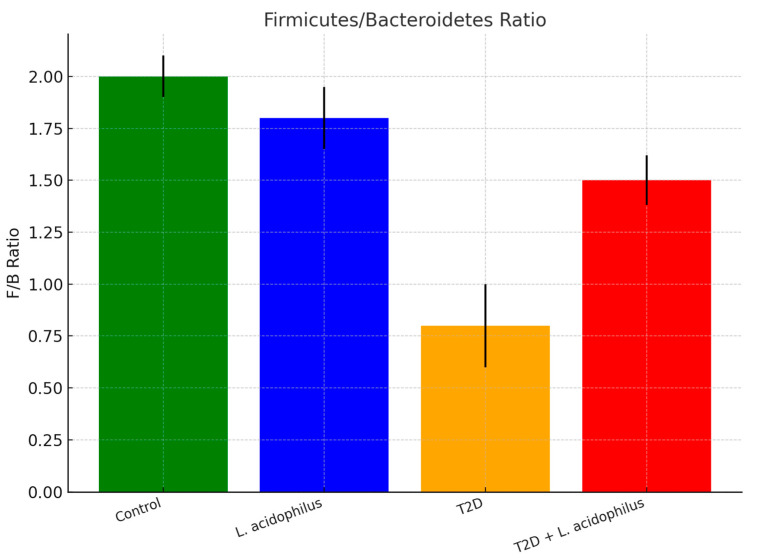
Firmicutes/Bacteroidetes ratio.

**Figure 9 biology-14-00122-f009:**
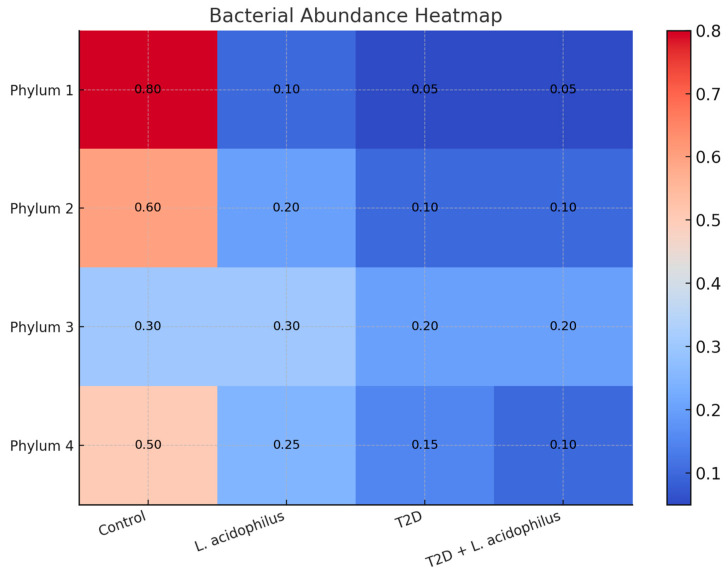
Bacterial abundance heatmap.

**Figure 10 biology-14-00122-f010:**
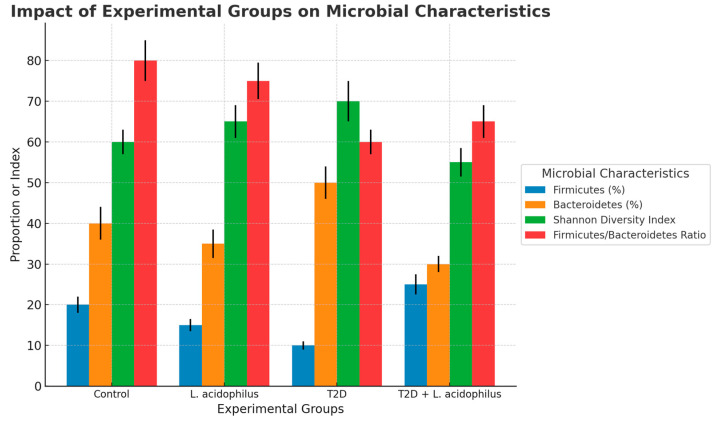
Microbial characteristics.

**Figure 11 biology-14-00122-f011:**
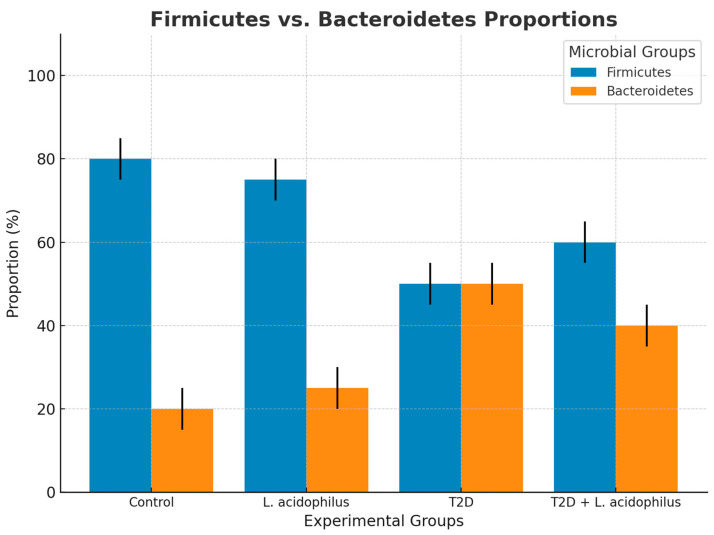
*Firmicutes*/*Bacteroidetes* proportions.

**Figure 12 biology-14-00122-f012:**
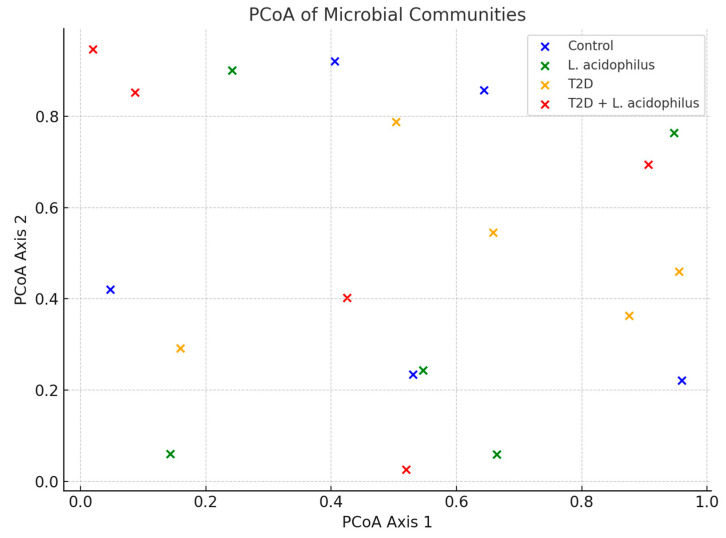
PCoA of microbial communities.

## Data Availability

All datasets generated or analyzed during this study are included in the manuscript.

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
