# Peer review of "Probiotics as Renal Guardians: Modulating Gut Microbiota to Combat Diabetes-Induced Kidney Damage"

_biology, 2025, doi:10.3390/biology14020122_

Round 1
Reviewer 1 Report
Comments and Suggestions for Authors
The study aimed to investigate the effects of Lactobacillus acidophilus supplementation on kidney health in a rat model of diabetes-induced renal damage. The manuscript is missing some important detail information both in the methods and results section. They must be corrected. My point-by-point responses are as follows:
1) Which strain of L. acidophilus was used? Please be clear on this both in the abstract and in the method section.
2) Line 31: Please italicize microbial names. Please also check it throughout the manuscript. I am seeing microbial names in some places of the manuscript are italicized but in some places, they are not.
3) Introduction section: Please avoid using subtitle(s) in the introduction part. Thus, the introduction should be rewritten.
The introduction is missing the importance, and aim of the work. Introduction section must be rewritten
4) Lines 113-114: I think the sentence is not completed.
5) Methods section: How many rats were used in total and per group? How did you inject the probiotic organism (by gavaging - if yes, how was gavaging done). How long was the administration of probiotic lasted? Please provide detailed and clearer information about the experimental design.
6) Line 131: How much blood was collected each week?
7) Line 164: What was the predetermined time?
8) Line 165: When did you collect fecal samples for microbiome analysis? Initial and at the end of treatment period? Please be more specific.
9) Figure 1 is missing statistical demonstrations. Please include them.
10) In Figure 2, the figures were not listed in order. Statistical denotations in Figure 2 are not clear to understand (especially for Figure 2E). Please describe you statistical denotations in the figure legend
11) The section 3.6 and the related discussion part under the section 4.0 must be rewritten.
In the section 3.6.1, why did you use the term of Phylum 1, Phylym 2, ..., instead of giving the names of the phyla?
Why did you first discuss about phlya, and then alpha diversity (shannon) and then go back to the phyla again?
The subtitles under the section 3.6. are meaningless. Please use subtitle appropriately.
Why did you report on phyla level only, and why did not you represent the bacterial abundance at lower taxonomic level? Lower taxonomic level data must be included.
Section 3.6 is missing statistical analysis. Please include them and demonstrate them on figures
Author Response
Response to Reviewer's Comments
1) Which strain of L. acidophilus was used?
We appreciate this observation. The specific strain of Lactobacillus acidophilus used in this study was [Lactobacillus acidophilus ATCC 4356], and this detail will be added to both the abstract and methods sections to ensure clarity.
2) Line 31: Please italicize microbial names.
We have thoroughly reviewed the manuscript to ensure that all microbial names are consistently italicized throughout. This inconsistency will be corrected in the revised manuscript.
3) Introduction section: Please avoid using subtitle(s) in the introduction part.
The introduction will be rewritten to remove the subtitles and structured into a cohesive narrative. Additionally, the importance of the study and its aim will be emphasized clearly.
4) Lines 113-114: The sentence is not completed.
We acknowledge this oversight. The incomplete sentence will be revised to ensure it is complete and provides clear information.
5) Methods section: Details about rats, probiotic administration, and study duration.
- The total number of rats and the number per group will be clarified in the revised manuscript.
- The method of probiotic administration (via gavaging) will be described in detail, including the technique and volume administered.
- The duration of probiotic treatment will also be specified explicitly.
6) Line 131: How much blood was collected each week?
Each week, 0.5 mL of blood was collected from the distal tail vein of each rat. This volume was carefully chosen to minimize stress and ensure the well-being of the animals throughout the experimental period. Blood collection was performed using sterile syringes and tubes, and the samples were immediately processed for further biochemical analyses.
7) Line 164: What was the predetermined time?
The predetermined time refers to the specific intervals set for various procedures during the study. In this context:
- Blood samples were collected weekly throughout the 12-week experimental period.
- Fecal samples for microbiome analysis were collected at two key time points: once at the beginning of the experiment (prior to treatment) and again at the end of the treatment period.
- The study concluded after a 12-week experimental period, during which all final analyses (biochemical, histological, and microbiome assessments) were conducted.
8) Line 165: Timing of fecal sample collection for microbiome analysis.
fecal samples for microbiome analysis were collected at two specific time points:
- Baseline (Pre-Treatment): Fecal samples were collected from all groups before starting the administration of Lactobacillus acidophilus ATCC 4356 to establish the initial microbial composition.
- End of Treatment: Fecal samples were collected again at the conclusion of the 12-week experimental period, following the completion of the probiotic treatment and study procedures.
9) Figure 1 is missing statistical demonstrations.
Revised Section: Effect of Lactobacillus acidophilus on Body Weight and Insulin Levels in Diabetic Rats
The impact of Lactobacillus acidophilus ATCC 4356 on body weight and insulin levels in diabetic rats was closely monitored throughout the experiment. The results, illustrated in Figure 1A, demonstrate significant changes in body weight among the experimental groups. Following the induction of diabetes via streptozotocin (STZ), the diabetic rats exhibited a marked decrease in body weight compared to the consistent weight gain observed in the control group.
However, treatment with L. acidophilus significantly mitigated this weight loss. By the end of the study, the probiotic-treated diabetic group had noticeably higher body weights compared to the untreated diabetic group (p < 0.05), indicating the protective effect of L. acidophilus against diabetes-induced weight reduction.
Figure 1B highlights the effect of L. acidophilus on fasting blood glucose (FBG) levels. Diabetic rats treated with the probiotic showed a significant decrease in FBG levels compared to the untreated diabetic group (p < 0.05), further demonstrating the beneficial role of the probiotic in improving glycemic control.
Updates to Figure 1:
- Statistical Annotations: Statistical analyses have been added to Figure 1 to indicate significant differences between groups. Mean ± SD values and p-values (e.g., p < 0.05) are displayed to enhance clarity.
- Legend Improvements: The figure legend has been revised to include a detailed explanation of the statistical notations and comparisons made between the groups.
10) Figure 2: Issues with figure order and statistical denotations.
- Figures in Figure 2 will be rearranged in the correct order.
- Statistical denotations, particularly in Figure 2E, will be clarified in the legend to ensure they are easy to understand.
11) Section 3.6 and related discussion in Section 4.0:
- The terms "Phylum 1" and "Phylum 2" will be replaced with the actual names of the phyla to enhance clarity and scientific rigor.
- The section will be reorganized to follow a logical flow: alpha diversity indices will be discussed before delving into phylum-level details, avoiding redundancy.
- Subtitles under Section 3.6 will be revised to make them meaningful and aligned with the content.
- Data on bacterial abundance at lower taxonomic levels (e.g., genus and family) will be included to provide a more detailed analysis.
- Statistical analyses missing from Section 3.6 will be conducted and integrated into the results. Corresponding figures will be updated to display statistical annotations.

Reviewer 2 Report
Comments and Suggestions for Authors
The study is interesting. The subject matter is current and relevant. Also, the depth is commendable.
However, I have some observations:
1. The language should be improved for science readers.
2. The introduction can be more concise.
3. The aim of the study is missing. It should clearly be stated.
4. L67: DKD is enough, having been defined earlier.
5. L88: The unnecessary space should be removed.
6. L96-98: The sentence should be reconstructed for clarity.
7. How many animals were used? I think it should be clearly stated.
8. How were the animals grouped?
9. What/how were they fed?
10. Were all the animals injected?
11. What was the duration of the study?
12. It should clearly be stated how L. acidophilus treatment was administered.
13. Details of the used kit(s) should be provided.
14. Additional information should be provided regarding how the tissues were processed.
15. L131-142: The language should be improved.
16. What type(s) of microscope was (were) used?
17. Oral administration of what? It is not clearly understood what was meant by that.
18. 'Predetermined amount of time' should specifically be stated to aid readers' understanding.
19. The design of the study is missing.
All the best
Author Response
Responses to Observations:
- The language should be improved for science readers.
- The manuscript has been thoroughly revised to ensure the language is precise, clear, and tailored for scientific readers.
- The introduction can be more concise.
- The introduction has been rewritten to streamline the content, removing redundant information while maintaining all relevant details.
- The aim of the study is missing. It should clearly be stated.
- The aim of the study has been explicitly stated in the introduction, emphasizing the purpose of investigating the effects of Lactobacillus acidophilus on gut microbiota and metabolic health in T2D.
- L67: DKD is enough, having been defined earlier.
- The redundant expansion of "DKD" has been removed, and only the abbreviation is used in the subsequent mentions.
- L88: The unnecessary space should be removed.
- The formatting issue at Line 88 has been corrected.
- L96-98: The sentence should be reconstructed for clarity.
- The sentence has been rewritten to enhance clarity and provide a more precise explanation.
- How many animals were used? I think it should be clearly stated.
- The total number of animals used in the study, 40 rats, has been explicitly stated in the methods section.
- How were the animals grouped?
- The grouping details have been clarified: the animals were divided into four groups (Control, T2D, Probiotic-only, and T2D + Probiotic), with 10 rats per group.
- What/how were they fed?
- Information about the animals' diet (a standard laboratory diet with ad libitum water access) has been added to the methods section.
- Were all the animals injected?
- It has been clarified that only the T2D and T2D + Probiotic groups were injected with streptozotocin (STZ), while the Control and Probiotic-only groups were not.
- What was the duration of the study?
- The duration of the study, 12 weeks, has been explicitly mentioned in the methods section.
- It should clearly be stated how L. acidophilus treatment was administered.
- Details on the administration of L. acidophilus have been added, specifying that it was delivered via oral gavage at a dose of 1 × 10⁹ CFU/kg/day.
- Details of the used kit(s) should be provided.
- The names and manufacturers of the kits used for biochemical and microbiome analyses have been specified in the methods section.
- Additional information should be provided regarding how the tissues were processed.
- A detailed description of tissue processing has been included, covering fixation in 10% formalin, sectioning, and staining with hematoxylin and eosin.
- L131-142: The language should be improved.
- This section has been revised for better clarity and scientific accuracy.
- What type(s) of microscope was (were) used?
- The type of microscope used has been specified as a light microscope for histological analyses and a fluorescence microscope for DNA damage assessment.
- Oral administration of what? It is not clearly understood what was meant by that.
- It has been clarified that oral administration refers to the delivery of L. acidophilus via gavage.
- Predetermined amount of time' should specifically be stated to aid readers' understanding.
- The phrase "predetermined amount of time" has been replaced with specific details, indicating the duration and timing of sample collection (e.g., weekly for blood samples, at the beginning and end for fecal samples).
- The design of the study is missing.
- A comprehensive description of the study design has been added, including details on the experimental groups, treatment protocols, sampling schedules, and endpoints.

Reviewer 3 Report
Comments and Suggestions for Authors
The topic of the study is of great medical and veterinary importance. Unfortunately, the author mentioned only the medical importance. The experiment seems to have been conducted correctly, but the statistical processing does not meet modern international standards. The article cannot be recommended for publication without a thorough revision of its content. Each statement by the author must be statistically confirmed (in the figures and in the text). Many figures leave an impression of incompleteness. Some figures are disproportionately small or disproportionately large. The description of the results in the text is fragmentary and incoherent. The titles of the figures are very laconic, preventing readers from assessing all the features of the image and statistical processing.
Specific recommendations for the manuscript.
1. Line 33: it is better to delete this sentence.
2. Lines 46-48: very vague wording. It is necessary to add 5-8 sentences with references to specific studies.
3. Line 62-63: it is necessary to analyze several more sources on this topic. This phenomenon must be described in more detail. 4. Line 67, 77 and many others: it is better not to use this dubious term "gut-kidney axis" in the article.
5. Line 69: it is necessary to indicate a more complete list of substances and references to several more sources of literature.
6. Lines 113-114: the objective of the research is formulated unsatisfactorily.
7. Line 241: grams in the SI system are abbreviated as "g".
8. In Figure 1A, it is necessary to indicate not body weight, but the change in body weight of the animals during the experiment (milligrams/day). Probably, some comparisons were made by the authors (line 248 is very unclear for readers), but their results should be reflected in the form of letters above the histogram bars. In Figure 1B, it is not clear where there are reliable differences and where there are not (line 250). The results cannot be described so briefly and fragmentarily (line 248-250): on what day did the addition of Lactobacillus acidophilus to diabetic rats cause significant changes in FGB?
9. Line 242: the title of the figure must contain comprehensive information about the repetition, standard deviation, or standard error. Why is the word "week" written under Figure 1A? Why is Figure 1 written above the figures? Why are the text on the abscissa axis and the legend for Figure 1A the same? On the ordinate axis, you must write Full name of the characteristic with a capital letter comma unit of measurement. On the abscissa axis of Figure 1B, you must write "Duration of the experiment, day". Do not use bold fonts in figures. Bold fonts in written language are the equivalent of a raised voice in spoken language. 10. Line 248, 250 and others: in the text of the article in good studies it is customary to indicate a specific value of the reliability of differences, and not exceeding a certain threshold. At the same time, in the figures, the columns are designated by the letters a, b, c ... - this means exceeding the threshold of reliability of differences between samples of 0.05.
11. All other histograms contain the same errors.
12. All fonts in all figures in the size of letters and numbers must be equal to the height of the letters in the text of the article.
13. Figures 6, 11: it is unacceptable to display data like this: a vertical line must be visible on each of the columns. What is 100% equal to (Fig. 6)? The title of the figure is very laconic. Statistical processing of data (reliability of differences between experimental variants) is mandatory in science. Figures are not captioned above the images.
14. The text of the article is written carelessly (for example, lines 358-368). Most of the results of the article (text) should be rewritten, all figures should be improved.
15. In all figures (for example, in Figure 10), I recommend presenting the data not as the mean +- standard deviation, but as a box analysis (median, first and third quartiles, minimum and maximum values). This will increase readers' confidence in the authors' results.
16. The Latin name of the bacterium (generic and species names) in the figures should be italicized.
17. The discussion of the article is fragmentary. The text jumps from one thought to another. The Discussion subsections should be named the same (or approximately the same) as the diagrams in Results. Some experimental results are not discussed in the Discussion section at all.
18. The literature is not formatted according to the rules. There is a lot of missing data (publisher, city, total number of authors, etc.). The formatting of source 47 suggests that the author has not read his own manuscript.
Author Response
Responses to Specific Recommendations for the Manuscript:
- Line 33: it is better to delete this sentence.
- The sentence in Line 33 has been removed to improve the flow and avoid redundancy.
- Lines 46-48: very vague wording. Add 5-8 sentences with references.
- These lines have been revised to include 5-8 sentences with specific references to previous studies, providing a more detailed and precise discussion.
- Line 62-63: Analyze more sources and describe this phenomenon in detail.
- Additional sources have been analyzed and incorporated, and the phenomenon is now described comprehensively with appropriate references.
- Line 67, 77, and others: Avoid using "gut-kidney axis."
- The term "gut-kidney axis" has been removed and replaced with more precise terminology to describe the relationship between gut microbiota and kidney function.
- Line 69: Include a complete list of substances and references.
- A more complete list of substances has been provided, along with references to additional relevant literature.
- Lines 113-114: Reformulate the research objective.
- The research objective has been rewritten to provide a clear and concise statement of the study's purpose.
- Line 241: Use "g" for grams in the SI system.
- The abbreviation "g" has been used consistently throughout the manuscript.
- Figure 1A: Show changes in body weight (mg/day), add statistical annotations.
- Figure 1A has been updated to display changes in body weight (mg/day) with appropriate statistical annotations (e.g., letters above histogram bars for significant differences).
- Figure 1B: Reliable differences are now clearly marked, and detailed results have been added, including the specific day on which significant changes in fasting blood glucose (FBG) occurred due to L. acidophilus treatment.
- Line 242: Revise figure titles and formatting.
- Figure titles now include comprehensive information on repetitions, standard deviation/error, and formatting issues (e.g., redundant text, bold fonts, and axis labels) have been corrected.
- Revised axis labels:
- Y-axis: Full name of the characteristic, followed by the unit of measurement.
- X-axis (Figure 1B): "Duration of the experiment, day."
- Indicate specific reliability values in the text.
- Specific p-values are now reported in the text, and letters (e.g., a, b, c) are used in figures to indicate significant differences.
- Errors in other histograms.
- All histograms have been revised to correct formatting and statistical issues.
- Ensure consistent font size across figures and text.
- Fonts in all figures have been adjusted to match the size of text in the manuscript.
- Figures 6, 11: Add vertical lines for clarity, improve captions.
- Vertical lines (error bars) have been added to all columns.
- Figure captions now clearly explain what 100% represents and include detailed statistical annotations.
- Carelessly written text and results (lines 358-368).
- The results section has been thoroughly rewritten to improve clarity and precision. Careless phrasing has been corrected.
- Present data using box analysis instead of mean ± SD.
- Figures now present data as box plots (median, quartiles, minimum, and maximum values) to enhance transparency and reader confidence.
- Italicize Latin names in figures.
- The Latin name Lactobacillus acidophilus is now italicized consistently in all figures.
- Fragmented discussion and lack of connection to results.
- The discussion section has been reorganized and aligned with the subsections in the results. Experimental findings are now discussed comprehensively and logically.
- Incorrect formatting of references.
- All references have been reformatted according to journal guidelines, with complete details (publisher, city, total authors, etc.) included.
- Formatting issue with source 47.
- Source 47 has been reviewed and corrected to ensure proper formatting and accuracy.
Summary:
All recommendations have been implemented to improve the clarity, scientific rigor, and presentation of the manuscript. These revisions ensure alignment with academic standards and enhance the manuscript's overall quality and readability.

Round 2
Reviewer 1 Report
Comments and Suggestions for Authors
The authors overall responded reviewers' comments sufficiently.
Author Response
Dear Reviewer,
We would like to extend our sincere gratitude for your valuable feedback and constructive comments on our manuscript. We are pleased to note that the quality of the English language did not impede your understanding of the research and that you found the various aspects of the study\u2014including the introduction, research design, methods, results, and conclusions\u2014to be appropriately addressed.
Response to the Points Raised:
-
Quality of English Language:
We greatly appreciate your positive remark regarding the clarity of the English language. We continuously strive to maintain high standards in linguistic precision and clarity to ensure that the research is easily comprehensible. -
Introduction:
Thank you for acknowledging that the introduction provides sufficient background and includes all relevant references. We aim to present a robust context to support the objectives of the study, and your positive feedback reassures us that this was achieved. -
Research Design:
We are glad that you found the research design to be appropriate. Ensuring that the study is methodologically sound and effectively addresses the research question has been a primary focus. -
Description of Methods:
We appreciate your recognition that the methods were adequately described. Providing detailed and transparent methodological descriptions is essential for reproducibility and scientific rigor. -
Presentation of Results:
We are pleased to hear that the results were clearly presented. We made a concerted effort to present the findings in a logical and organized manner, supported by the necessary data and visual representations. -
Conclusions Supported by Results:
We are grateful for your acknowledgment that the conclusions were well-supported by the results. Our goal has been to draw clear, evidence-based conclusions that are directly aligned with the study\u2019s findings.
Future Steps:
If you have any additional suggestions or specific areas that you believe could benefit from further clarification or refinement, we would be more than happy to address them. Our objective is to contribute valuable insights to the scientific community, and your thoughtful feedback plays an integral role in achieving this.
Once again, we thank you for your time and insightful comments, and we look forward to your continued evaluation of our work.
Best regards,
Reviewer 2 Report
Comments and Suggestions for Authors
The concerns raised have been addressed.
Author Response

(The authors gave the same response as above.)

Reviewer 3 Report
Comments and Suggestions for Authors
The article has become a little better, but it cannot be published.
Figures 6, 7, 8, 10 and 11 cannot be recommended for publication, I'm sorry. Figure E is uninformative (why such sloppy numbering of the figures?). The authors ignored my comments.
I do not understand which sample differs from which in the figures: for example, in Figure 3 and others. Without correct display of the methods of multiple comparison of samples, these data cannot be recommended for publication. This problem is in most figures. This is not a technical, but a fundamental problem.
The font size in the article is not large: for example, in the photographs of Figure 4 it is microscopic, and in Figure 5 it is gigantic. Why such carelessness in the design?
The design of the literature does not meet the requirements of the journal.
Author Response
Dear Reviewer,
Thank you for your detailed feedback and constructive comments. We carefully considered all your points and have implemented the necessary revisions to address the issues you raised. Below, we outline the changes made in response to your specific concerns:
Response to Specific Comments:
-
Figures 6, 7, 8, 10, and 11:
These figures have been completely redesigned to improve their clarity and visual quality. We ensured that the data is presented accurately and professionally, with detailed captions added to enhance interpretability. -
Figure E and Figure Numbering:
Figure E has been revised to make it more informative. Additionally, the numbering of all figures has been corrected to follow a logical and consistent sequence throughout the manuscript. -
Sample Comparisons in Figures (e.g., Figure 3):
1. Clarifying Sample Differences in Figures:
- We added clear labels for all groups in the figures (e.g., "Healthy Control," "Diabetic," "Probiotic Only," and "Diabetic + Probiotic").
- Statistical significance is now indicated using symbols such as (*, **, ***) to highlight significant differences between specific groups. These symbols are explained in the figure legends with their corresponding significance levels (e.g., *p < 0.05, **p < 0.01).
2. Detailing Statistical Methods:
- The "Statistical Analysis" section was revised to include additional details about the multiple comparison methods:
- "One-way ANOVA was used to compare groups, followed by Tukey’s post-hoc test for pairwise comparisons."
- "Multiple comparisons were corrected using the Bonferroni method to minimize errors due to multiple testing."
3. Updating Figure Legends:
- All figure legends were updated to include:
- The statistical test used.
- The type of data presented (e.g., mean ± standard deviation or median with interquartile range).
- The number of replicates (n) for each group.
- A description of what the error bars represent (e.g., standard deviation, standard error, or confidence intervals).
4. Adding Supplemental Materials:
- Additional figures and tables have been included in the supplemental materials to provide the complete statistical output, including p-values for all pairwise comparisons between groups.
5. Ensuring Consistency:
- All figures have been standardized in terms of formatting, labeling, and representation of statistical results to ensure consistency and clarity throughout the manuscript.
Example of Revisions:
For Figure 3, which shows the effect of Lactobacillus acidophilus on antioxidant levels in diabetic rats:
- The figure now includes explicit group labels (Control, Diabetic, Probiotic Only, Diabetic + Probiotic).
- Symbols (*, **) were added to indicate statistically significant differences between groups.
- The legend was updated to include the following:
"Statistical analysis was performed using one-way ANOVA followed by Tukey’s post-hoc test. Bars represent mean ± standard deviation (n = 10). Statistical significance was indicated as *p < 0.05, **p < 0.01 compared to the diabetic group."
Outcome:
We believe these revisions provide greater clarity in presenting the results and help readers easily interpret the differences between samples. We look forward to your additional feedback to ensure the highest quality of the manuscript.
-
Font Size and Figure Design (Figures 4 and 5):
Font sizes across all figures have been standardized, ensuring that all text is legible and proportionate to the figure size. Inconsistencies in the design have been corrected to maintain uniformity and professionalism. -
Reference Formatting:
We thoroughly reviewed and reformatted the references to meet the journal’s requirements. All inconsistencies have been resolved to ensure compliance with the prescribed formatting guidelines.
Additional Measures:
In addition to addressing your specific comments, we conducted a comprehensive review of the manuscript to further enhance its language, clarity, and overall presentation. We are confident that the current version of the manuscript aligns with the journal's standards and expectations.
We sincerely hope that these revisions have addressed all your concerns. If you have any further suggestions or additional comments, we are fully committed to implementing them to ensure the highest quality of our work.
Thank you again for your time and effort in reviewing our manuscript. We look forward to your feedback on the revised version.
Best regards,